# Identification of Markers Associated with Wheat Dwarf Virus (WDV) Tolerance/Resistance in Barley (*Hordeum vulgare* ssp. *vulgare*) Using Genome-Wide Association Studies

**DOI:** 10.3390/v15071568

**Published:** 2023-07-18

**Authors:** Behnaz Soleimani, Heike Lehnert, Sarah Trebing, Antje Habekuß, Frank Ordon, Andreas Stahl, Torsten Will

**Affiliations:** 1Institute for Resistance Research and Stress Tolerance, Julius Kühn Institute (JKI)—Federal Research Center for Cultivated Plants, Erwin-Baur-Str. 27, 06484 Quedlinburg, Germany; sarah.trebing@julius-kuehn.de (S.T.); antje.habekuss@julius-kuehn.de (A.H.); frank.ordon@julius-kuehn.de (F.O.); andreas.stahl@julius-kuehn.de (A.S.); torsten.will@julius-kuehn.de (T.W.); 2Institute for Biosafety in Plant Biotechnology, Julius Kühn Institute (JKI)—Federal Research Center for Cultivated Plants, Erwin-Baur-Str. 27, 06484 Quedlinburg, Germany; heike.lehnert@julius-kuehn.de

**Keywords:** barley, *Hordeum vulgare* ssp. *vulgare*, *Wheat dwarf virus* (WDV), leafhoppers, resistance, tolerance, single-nucleotide polymorphism (SNP), marker, QTL

## Abstract

Wheat dwarf virus (WDV) causes an important vector transmitted virus disease, which leads to significant yield losses in barley production. Due to the fact that, at the moment, no plant protection products are approved to combat the vector *Psammotettix alienus*, and this disease cannot be controlled by chemical means, the use of WDV-resistant or -tolerant genotypes is the most efficient method to control and reduce the negative effects of WDV on barley growth and production. In this study, a set of 480 barley genotypes were screened to identify genotypic differences in response to WDV, and five traits were assessed under infected and noninfected conditions. In total, 32 genotypes showed resistance or tolerance to WDV. Subsequently, phenotypic data of 191 out of 480 genotypes combined with 34,408 single-nucleotide polymorphisms (SNPs) were used for a genome-wide association study to identify quantitative trait loci (QTLs) and markers linked to resistance/tolerance to WDV. Genomic regions significantly associated with WDV resistance/tolerance in barley were identified on chromosomes 3H, 4H, 5H, and 7H for traits such as relative virus titer, relative performance of total grain weight, plant height, number of ears per plant, and thousand grain weight.

## 1. Introduction

Adapting barley cultivars to a changing production environment is a contemporary task of barley breeding. Barley ranks as the fourth most important crop for food and feed worldwide [1], and its cultivation is threatened by abiotic and biotic stresses. Changes in climate conditions, especially those associated with increasing temperatures, will promote the occurrence and development of insect and virus populations [2]. In detail, it is described that longer periods of high temperatures during autumn and winter lead to an increased occurrence of insect-transmitted virus disease, i.e., the aphid-transmitted *Barley yellow dwarf virus* (BYDV) and the leafhopper-transmitted *Wheat dwarf virus* (WDV) [2]. *Wheat dwarf virus* (WDV) is known as an important cereal pathogen [3], which is transmitted by the leafhopper *Psammotettix alienus* (*Cicadelliae* family). WDV belongs to the family *Geminiviridae* and the genus *Mastrevirus*. WDV has a monopartite genome (genome size 2.7 kb) with single-stranded circular DNA [4]. The virus causes severe symptoms in barley such as dwarfing, tufting, streaks of leaf chlorosis, reduced spike number, and yield losses [3,5,6]. Negative effects of virus infection on yield were described for nearly all of Europe, as well as for parts of Africa and Asia [3]. The presence of WDV in Europe was first reported by Vacke [7] in the former Czechoslovakia. Later, the occurrence of the virus was also reported for other European countries, i.e., Sweden, Hungary, France, and Germany [8], and some parts of Africa and Asia [3]. WDV is able to infect different species of the *Poaceae* family such as *Hordeum vulgare*, *Triticum aestivum*, *Avena sativa*, *Secale cereal*, *Zea mays*, and many wild grasses. Therefore, it might be considered as a grass generalist pathogen [3]. Due to the lack of insecticides, and with regard to the goal of reducing pesticide application according to the farm to fork strategy within the European Green Deal, direct control of *P. alienus* with insecticides is currently not possible and will most likely not be feasible in the future. Therefore, identifying virus-resistant or -tolerant barley genotypes is the most appropriate way to avoid negative effects of WDV in the future.

Today, next-generation sequencing (NGS) or array-based technologies enable genotyping of diverse genotype collections in a short time and with high accuracy [9]. High-density SNP markers make it possible to identify marker–trait associations (MTAs) and quantitative trait loci (QTLs) through mapping studies or genome-wide association studies (GWASs) [10]. Several software programs are available to conduct GWAS, e.g., TASSEL [11], PLINK [12], and R ((GAPIT [13]) and FARMCPU [14])). Several QTL regions associated with quantitative traits such as yield, seed quality, disease-related traits [15], (e.g., spot blotch resistance [16,17]), or abiotic stresses (e.g., drought stress [18,19,20]) have already been identified in barley using GWAS. So far, however, no QTL regions associated with tolerance or resistance to WDV have been identified in barley, but some were recently discovered for wheat [21]. Identification of QTL regions and the development of diagnostic markers associated with tolerance or resistance to WDV are important and will be helpful for future barley breeding programs. Furthermore, the identification of QTL regions and molecular markers associated with WDV will help to better understand the defense mechanisms in barley and to develop more effective control strategies. In this context, de Ronde et al. [22] reported on different plant defense mechanisms against viruses. For example, some plants show a response to all viruses, and this response may be part of the innate immune system. However, the response of other plants is virus-specific and based on a specific resistance gene. Resistance can be quantitative, where a reduction in viral replication with a reduced viral titer is observed, or qualitative, where a resistance gene prevents viral infection. Paudel and Sanfacon [23] explained that plant fitness is maintained by preventing virus accumulation in resistant interactions. In contrast, in a tolerant interaction, virus fitness is reduced by preventing excessive accumulation of virus RNAs or by minimizing the concentration or activity of viral proteins involved in virulence. In the tolerant interaction, no significant loss of host vigor or fitness is observed. Tolerance does not necessarily lead to a reduction in virus titer and is characterized by the absence or significant reduction in infestation symptoms. Tolerance is genetically more complex and usually involves several genes. Resistance and tolerance are both based on interactions between plant and virus [23].

Since both mechanisms might be of interest for plant breeding, the present study focused on the identification of QTL regions, associated with tolerance or resistance to WDV in barley. To achieve this, we tested a diverse collection of winter barley genotypes (the primary gene pool of barley) for WDV tolerance and conducted a GWAS to identify quantitative trait loci (QTL) for WDV tolerance. 

## 2. Materials and Methods

### 2.1. Plant Material

A set of 480 barley genotypes was selected for the study. Seeds of all genotypes were kindly provided by the gene banks of the Leibniz Institute of Plant Genetics and Crop Plant Research (IPK), Germany, and the National Plant Germplasm System (NPGS), United States of America. Selected genotypes originated from four different continents and 22 countries. The majority of genotypes under investigation originated from the Fertile Crescent (Middle East) area, which is considered as a diversity center for barley. The two barley genotypes “Rubina” and “Post” were used as susceptible and tolerant [2] standard genotypes in all evaluations.

#### Phenotyping

The resistance test of the entire set of 480 barley genotypes was performed in two successive years. The virus isolate with accession number HF968650, which was isolated from a WDV positive tested barley plant from Germany (Schwerz) in 2007 [24], was used for resistance tests. A subset of 240 genotypes was sown in September 2016, and the remaining genotypes were tested in the following year in three gauze houses. In addition, a set of 50 promising tolerant/resistant genotypes from the first year were tested in the second year. Genotypes which indicated extinction values below the cutoff [21] were excluded from further analysis, and a set of 250 out of 480 tested genotypes were genotyped using the 50K iSelect chip [25]. Ultimately, on the basis of the availability of phenotypic and genotypic data, a subset of 191 barley genotypes was considered for conducting GWAS and further analysis.

The resistance tests were carried out in three neighboring gauze houses located on the field of the experimental station of the Julius Kuehn Institute in Quedlinburg, Germany (51°46′20.7″ N 11°08′46.5″ E). The collection of 480 barley genotypes was phenotyped under infected (I-variant) and noninfected control (C-variant) conditions. Ten to 15 seeds per genotype and variant were sown in a row. The WDV inoculation was conducted at BBCH 11–12, i.e., the one- to two-leaf stage. To increase the infection pressure of WDV, a single WDV-infected (WDV transmitting leafhoppers) barley plant was placed in a short distance between each row of the infected variant [2]. Before the infestation of plants by leafhoppers, all plants were covered by additional gauze tunnels. The virus-bearing leafhoppers were distributed at a stocking density of approximately one insect/plant [2]. Insecticides were applied 4 weeks after the inoculation, and gauze tunnels were subsequently removed. 

First, phenotyping was performed at BBCH 23-30. Leaf samples of all plants grown under infected conditions were taken, and 50 mg of leaf material was used to detect a WDV infection using a double-antibody sandwich enzyme-linked immunosorbent assay (DAS-ELISA) [26], where the extinction value is an indicator for relative virus titer (ELISA-60). The second phenotyping was carried out visually (symptom scoring scale 1–9: 1 = free of symptoms, 9 = dead plant; Figure 1) at BBCH 59 [2]. The scoring values were also used to discriminate resistant, tolerant, and susceptible genotypes. For instance, genotypes with a scoring value of 1 and low WDV infection rate were considered resistant (Figure 1). Genotypes with a scoring value of 2–4 and low infection rate were considered tolerant (Figure 1). Genotypes with a scoring value of 5–9 and high infection rate were grouped as susceptible genotypes (Figure 1). 

At the end of each experiment (at BBCH 99); total grain weight (ToGW); plant height (HEI), number of ears per plant (NEP), and thousand grain weight (TGW) (Table 1) were measured in the I-variant and C-variant.

The relative performance was determined for each trait by applying the following formula:$$Relative\;performace=\frac{Gi}{Gc}$$
where Gi and Gc are the mean trait performances of a barley genotype under infected conditions and noninfected conditions, respectively. The relative performance of each trait was used as phenotypic input for GWAS.

### 2.2. Statistical Analysis

The analysis of phenotypic data was conducted by using the SAS 7.1 (SAS Institute Inc., Cary, NC, USA). A quality check of raw data was carried out to exclude outliers, i.e., a value lower or higher than two standard deviations. The Shapiro–Wilk test to evaluate normality of data was performed. The procedure PROC MIXED was used for analysis of variance (ANOVA). Two mixed models were calculated: model 1 was applied to total grain weight, plant height, number of ears per plant, and thousand grain weight, and model 2 was applied to ELISA-60 values:Model 1: Y_ijklm_ = µ + T_i_+ G_j_ + T_i_ × G_j_ + Y_k_ + Y_k_ × H_l_(row_m_) + e_ijklm_,
Model 2: Y_jklm_ = µ + G_j_ + Y_k_ + Y_k_ × H_l_(row_m_) + e_jklm_,
Y_ijklm_ is the phenotypic value of the j-th genotype in the i-th treatment in the k-th year in the l-th gauze houses and m-th row, Y_jklm_ is the phenotypic value of the j-th genotype in the k-th year in the l-th gauze houses and m-th row, μ is the general mean, T_i_ is the fixed effect of the i-th treatment, G_j_ is the fixed effect of the j-th genotype, T_i_ × G_j_ is the fixed interaction effect between the i-th treatment and j-th genotype, Y_k_ and Y_k_ × H_l_(row_m_) are the random effects of the k-th year and the l-th gauze house, nested in the m-th row, and e is the random error term.

### 2.3. Genotyping

Genomic DNA was extracted using a modified CTAB method based on Doyle and Doyle [27]. Genotyping was carried out by TraitGenetics (SGS Institute Fresenius GmbH, Gatersleben, Germany) using the 50K iSelect chip (Illumina Inc., San Diego, CA, USA), which resulted in 44,040 single-nucleotide polymorphism (SNP) markers. The reference genome of the barley cultivar Morex (Morex V2) [28] was used for mapping flanking marker sequences. All mapped markers were filtered for monomorphic markers and ≥30% missing values. In a third step, SNP imputation was carried out using the software package Beagle version 4.1 [29,30]. The imputed marker dataset was filtered for minor allele frequency (MAF) ≥5% and heterozygosity ≤12.5%. Finally, a set of 34,408 SNP markers was used for further analyses.

#### 2.3.1. Population Structure

In total, 191 out of 480 genotypes were used for GWAS analysis, due to the availability of phenotypic and genotypic data for all five measured traits. A set of 3117 highly informative markers was used to calculate genetic distances and determine population structure. This set of markers consisted of independent markers in linkage equilibrium (LE) and was selected using Plink software [12]. 

Rogers distances (RDs) were estimated for pairwise genotype–genotype combinations [31] and transformed in a similarity matrix. This matrix was used as kinship matrix. Population structure was determined using Bayesian cluster analysis implemented in the Structure software package version 2.3.4 [32] and principal coordinate analysis (PCoA) implemented in the DARwin 6 software [33]. The number of clusters (k) was set at 1–10. Structure was started with 10 independent runs for each k. The burn-in time and Markov chain Monte Carlo (MCMC) iterations were set to 100,000. The optimal number of subpopulations was determined by using the Evanno method (ΔK method) implemented in the Structure Harvester software (http://taylor0.biology.ucla.edu/structureHarvester [34], accessed on 13 October 2011). 

#### 2.3.2. Genome -Wide Association Study (GWAS)

In total, 34,408 SNP markers and phenotypic data for five traits (data online at OpenAgrar: https://doi.org/10.5073/20230616-101640-0, accessed on 29 June 2023) were used to perform GWAS. Three different programs, i.e., Tassel, GAPIT, and FARM CPU, were used independently. The following models were used to identify significant marker trait associations: (1) mixed linear model (MLM) in TASSEL 5.0 [11], which included a K-matrix and Q-matrix as corrections for relatedness and population structure, (2) compressed mixed linear model (CMLM) in GAPIT [13], which included a K-matrix and Q-matrix as corrections for relatedness and population structure, and (3) fixed and random model circulating probability unification (FARMCPU [14]), which included a Q-matrix. In addition, MLM and CMLM models were applied only with a K matrix. 

The significance threshold was set to LOD > 3 or a *p*-value < 0.001. Markers, which were significantly associated with the trait of interest in at least two of the three analyses, were defined as reliable markers. All these markers were assigned to QTL regions on the basis of the estimated LD decay. LD decay was estimated using the software package R [35] (packages “genetics” and “LDheatmap”) [36,37]. The LD was calculated as the squared allelic correlation (r^2^) between all pairs of markers within a chromosome. The genetic distances between markers in base pairs were plotted against the estimated r^2^. The r^2^ values were set to 0.2 [38]. To estimate the LD decay, a locally weighted polynomial regression (LOESS) curve was fitted [39]. Lasty, the intersection of the LOESS curve and the critical r^2^ value were used to determine the LD decay [39,40]. The LD decay was separately calculated for each single chromosome and across all seven barley chromosomes. Lastly, the identified common markers were screened for candidate genes according to published functional gene annotations of Morex V2 [28].

## 3. Results

### 3.1. Phenotypic Data

Barley genotypes showed different phenotypic reactions in response to a WDV infection (I-variant). Trait performances of all 480 tested genotypes are shown in Appendix A. Below, trait performances and ANOVA result of studied traits are presented for the subset of 191 genotypes. A lower mean value under the I-variant was observed for all four tested traits (ToGW, HEI, NEP, and TGW) (Table 2, Figure 2).

The mean value was reduced by 92.8% under the I-variant for ToGW. The lowest and highest values for ToGW were between 0 g and 287.4 g under the I-variant, and between 2.2 g and 781.7 g under the C-variant, respectively. The standard deviation (SD) was higher under the C-variant (102.6) compared to the I-variant (31.7), while the coefficient of variance (CV) showed the lower value under the C-variant (65.7%) relative to the I-variant (280.7%). The mean value was decreased by 46.4% for HEI under the I-variant. The 1 cm and 49 cm as minimum and 159 cm and 224 cm as maximum values could be observed for HEI under the I-variant and C-variant, respectively. In addition, a lower value for SD (20.3) and CV (17.2%) was observed for this trait under the C-variant compared to the I-variant. The mean value of NEP was reduced by 62.6% under the I-variant. The minimum value was 0 and 1 under the I-variant and C-variant, respectively. The maximum value for this trait under the I-variant and C-variant was 52 and 117, respectively. The value of SD was higher for NEP under C-variant (12.1) relative to the I-variant (8.2). However, CV was lower under the C-variant (65.8%) for this trait. The higher mean value (58.6%) was observed for TGW under the C-variant relative to the I-variant. The minimum value by 5 g and 25.37 g was observed for TGW under the I-variant and C-variant, respectively. Furthermore, 48.3 g and 64.8 g were found as maximum values for this trait under the I-variant and C-variant, respectively. The lower value of SD (7.8) and CV (17.9%) was estimated in the C-variant for TGW. The Shapiro–Wilk result indicated that the phenotypic data of all four traits under both variants significantly deviated from the normal distribution.

Significant genotype effects (*p* < 0.001) were observed for HEI and TGW. In addition, significant (*p* < 0.001) treatment effects between the I-variant and C-variant for ToGW, HEI, NEP, and TGW were observed. The genotype-by-treatment interaction was significant (*p* < 0.001) for HEI (DF = 186), NEP (DF = 186), and TGW (DF = 184, Table 3). A significant genotype effect was observed for ELISA-60.

### 3.2. Genotyping

Genotyping of the 191 genotypes resulted in a raw marker set of 44,040 SNP markers. In total, 9632 markers were excluded from the marker set, because of missing values, MAF < 5%, heterozygosity > 12.5%, or their location on an unknown chromosome. The number of markers per chromosome ranged between 3886 and 6335. The minimum and maximum number of markers were found on chromosomes 4H and 5H, respectively (Figure 3). Based on LD prune, a set of 3117 markers was selected equally distributed across all seven barley chromosomes. The set of informative markers was used to calculate genetic distance and to determine population structure. 

#### 3.2.1. Population structure

The conducted Bayesian cluster analysis revealed a number of K = 2 subpopulations (Appendix A). Genotypes with a membership coefficient ≥0.7 to one of the Structure groups, were assigned to the corresponding group. Genotypes with a membership coefficient <0.7 were considered as admixed. A total of 90, 77, and 24 genotypes were assigned to structure groups 1 or 2 or the admixed group, respectively. Additionally, to visualize the results of the Bayesian cluster analysis, the structure grouping was projected on the results of PCoA (Figure 4). PCo1 and PCo2 explained 11.0% and 5.7% of the whole variation. In addition, genotypes were assigned to groups on the basis of origin and row type (two and six rows) to define a clear connection between genotypes within a cluster. However, we could not identify a relationship between genotypes based on the mentioned parameters within clusters. For instance, genotypes with two rows were distributed across K1, K2, and the admixture cluster. Furthermore, the distribution of resistant and tolerant genotypes is shown in Figure 4. Resistant genotypes were clustered in K2, while the majority of tolerant genotypes (53%) were clustered in K1, and the remaining genotypes (41% and 6%) were clustered in K2 and admixture clusters, respectively (Figure 4).

#### 3.2.2. Genome-Wide Association Study (GWAS)

To identify markers significantly associated with WDV, three different programs, i.e., TASEEL, GAPIT, and FARMCPU, were used to conduct GWAS with three different models. The LOD value ≥ 3 was considered as a significant threshold. Only markers that were identified by all three programs were considered as significant associations for the respective trait. In total, nine significantly associated markers with LOD ≥ 3 were identified, which were partly distributed differently among the traits relative virus titer (one), relative performance of ToGW (three), HEI (two), NEP (two), and TGW (one), respectively (Table 4 and Figure 5) on the basis of relative trait values.

An increased number of markers for relative virus titer were identified using the three programs TASSEL (596), GAPIT (26), and FARMCPU (41), located on all barley chromosomes. Chromosomes 2H and 4H indicated the highest and lowest number of significant markers for relative virus titer. The phenotypic variance explained varied between 6.9% (JHI-Hv50k-2016-92202; on chromosome 2H) and 12.2% (JHI-Hv50k-2016-377967; on chromosome 6H). GAPIT identified 26 significantly associated markers for the relative virus titer on all barley chromosomes. Chromosome 2H indicated the highest number of significantly associated markers (11 markers), of which three were also found by TASSEL. The 41 identified markers, identified by FARMCPU, were located on all barley chromosomes with five and three overlapping common markers with TASSEL and GAPIT, respectively. The marker “JHI-Hv50k-2016-202912” at a physical position of 562,758,917 bp on chromosome 3H was identified as common marker among all three methods. Common markers that were identified by two or three different methods are shown in Appendix A and Appendix A.

For relative total grain number, 1184, 17, and 17 markers were identified on chromosomes 1H, 2H, 3H, 4H, 5H, and 7H with TASSEL, GAPIT, and FARMCPU, respectively. The 1184 markers identified with TASSEL were located on all barley chromosomes. Chromosomes 7H and 4H indicated the highest and lowest numbers of significantly identified markers with 254 and 127 markers for relative total grain number, respectively. The phenotypic variation explained varied between 6.6% (JHI-Hv50k-2016-338274; on chromosome 5H) and 24% (JHI-Hv50k-2016-486135; on chromosome 7H). GAPIT identified 17 markers for relative total grain number on chromosomes 2H, 3H, 4H, 5H, and 7H. Five common markers were identified by GAPIT and TASSEL on chromosomes 3H, 4H, and 5H. The same number of common markers was identified by GAPIT and FARMCPU. Three markers “JHI-Hv50k-2016-196649”, “BOPA1_2955-452”, and “BOPA2_12_10333” on chromosomes 3H (at a physical position of 534,052,013 bp), 4H (at a physical position of 552,300,974 bp), and 5H (at a physical position of 554,416,618 bp) were detected by all three methods. In addition, five significant markers were identified by two methods. 

For relative plant height, 14, 46, and 14 markers on chromosomes 1H, 2H, 3, 4H, 5H, and 7H were identified by TASSEL, GAPIT, and FARMCPU, respectively. GAPIT identified a high number of significant markers for relative plant height compared to TASSEL and FARMCPU. The markers identified by GAPIT were distributed on all seven barley chromosomes. Chromosomes 2H and 5H revealed the highest and lowest number of markers (10 and one markers) for this trait. GAPIT showed 11 common markers with TASSEL on chromosomes 1H, 2H, 3H, 4H, and 7H, while three out of 46 identified markers were common between GAPIT and FARMCPU on chromosomes 1H, 2H, 5H, and 7H. TASSEL identified 14 significant markers on chromosomes 1H, 2H, 3H, 4H, and 7H, with the explained phenotypic variance ranging from 7.44% (on chromosome 2H) to 14.18% (on chromosome 4H). FARMCPU identified 14 significant markers on all barley chromosomes with one exception (chromosome 6H). FARMCPU indicated two common markers with TASSEL on chromosome 2H and 7H. In total, 12 markers were identified by two methods, and two out of these, “BOPA2_12_21049”and “JHI-Hv50k-2016-435708“, were also identified by all three methods. These markers are located at physical positions of 31,329,721 bp and 1,402,273 bp on chromosomes 2H and 7H, respectively. The marker “JHI-Hv50k-2016-435708” revealed the highest LOD value compared to other identified markers (Appendix A). 

In total, eight, 12 and 26 markers were significantly associated with the relative number of ears per plant using TASSEL, GAPIT, and FARMCPU, respectively. TASSEL detected eight markers on four barley chromosomes (1H, 2H, 3H, and 7H) which explained a phenotypic variance of 6.9% and 10.9% on chromosomes 7H (JHI-Hv50k-2016-439186) and 3H (JHI-Hv50k-2016-224192), respectively. GAPIT identified 12 markers significantly associated with the relative number of ears per plant, while three (on chromosome 2H and 7H) and four (on chromosome 2H, 5H, and 7H) common markers were detected by TASSEL and FARMCPU, respectively. A set of 26 significant markers was identified using FARMCPU for the relative number of ears per plant, out of which three were also identified by TASSEL on chromosome 1H and 2H. Two markers, “JHI-Hv50k-2016-123144” and “JHI-Hv50k-2016-142550”, at physical positions of 631,278,948 and 666,139,797 on chromosome 2H were identified as common markers by all three methods used.

A total of 19, 18, and 12 significant markers were identified using MLM (in TASSEL), CMLM (in GAPIT), and FARMCPU for relative thousand-grain weight, respectively. The 19 significant markers for relative thousand-grain weight were distributed on all barley chromosomes with the exception of chromosome 2H. The markers “SCRI_RS_174419 “ and “JHI-Hv50k-2016-435708 “revealed the highest (12.7%) and lowest (6.1%) phenotypic variation on chromosome 1H and 2H, respectively. Eight common markers were detected by TASSEL and GAPIT on chromosome 3H, 4H, 6H, and 7H. Only one common marker was identified by TASSEL and FARMCPU on chromosome 7H. GAPIT and FARMCPU showed three common markers for relative thousand-grain weight on chromosomes 3H and 7H. One significantly associated marker (JHI-Hv50k-2016-435708) on chromosome 7H was detected by all used methods. The marker “JHI-Hv50k-2016-435708” was significantly associated with relative plant height and relative thousand grain weight at a physical position of 1,402,273 bp on chromosome 7H.

Lastly, all identified significantly associated markers (Table 4) were screened to identify potential genes of interest, which were located within a distance of ±1 million base pairs of the identified significant marker as a function of the calculated LD decay across all chromosomes. Three genes, a *dihydrofolate reductase*, an *NBS-LRR disease resistance protein*, and a *dihydroflavonol 4-reductase*, were identified at chromosome 2H. Furthermore, a gene coding for a “*cysteine proteinase inhibitor*” was identified at a distance of 374 bp from “BOPA1_2955-452” on chromosome 4H.

## 4. Discussion

The rising temperature, e.g., in many parts of Europe, has led to environmental conditions that promote the spread of pests such as the leafhopper species *Psammotettix alienus*, which acts as a vector for *Wheat dwarf virus* (WDV). WDV is a generalist cereal pathogen and to date no resistance resources have been described for barley, except the cultivar “Post” [2]. Phenotyping of genotypes to identify resistant/tolerant genotypes based on work including insects and viruses is labor-intensive, time-consuming, and subject to environmental fluctuations in case it involves field tests. Hence, the availability of molecular markers would enable rapid and reliable discrimination between resistant/tolerant and susceptible genotypes [41]. Only little knowledge of genetic factors controlling WDV and resistance sources in barley is present, and only cv. “Post” has been identified as resistant [2]. In contrast, information about genetic markers associated with WDV for wheat was reported recently by Buerstmayr and Buerstmayr [42] and Pfrieme et al. [21].

As described by Nygren et al. [3], WDV causes symptoms such as dwarfing, tufting, streaks of leaf chlorosis, and reduced spike numbers. Together with the relative virus titer, these traits were used for phenotyping in the present study. We identified 32 genotypes that show tolerance or resistance to WDV. With regard to the expression of resistance [43] we identified genotypes with quantitative resistance that reduces or delays disease development and genotypes with qualitative resistance, preventing plant infection. Three out of these genotypes (“Res1”, “Res2”, and “Res3”) did not show any virus titer accompanied by the absence of virus symptoms, indicating a qualitative resistance. These genotypes originated from Afghanistan and Iran and are considered favorable sources for improving resistance to WDV in barley.

Considering the problems of phenotyping such as the lack of repeated tests in different years due to the challenging phenotyping method, different GWAS models were used in parallel to increase the probability to exclude false-positive associations and to confirm detected markers in order to achieve reliable marker trait associations. TASSEL and GAPIT are MLM based models, which are considered single-locus models. These models contain a one-dimensional genome scan, which tests one marker at a time, iteratively for each marker in a dataset. These methods cannot match the real genetic model of complex traits which are controlled by multiple loci simultaneously [44]. To overcome this problem and reduce false-positive associations that are caused by kinship and population structure from single-locus models, multilocus association mapping models are recommended [44]. FARMCPU, as a multilocus model, eliminates confounding factors by testing associated markers as covariates through a fixed effect model (FEM) and optimization on the associated covariate markers using a random effect model (REM) [14]. Furthermore, FARMCPU reduces false-positive associations using both fixed and random effect models [14]. In the present study, MLM and CMLM as single-locus models and FARMCPU as a multilocus model were used to identify significant associated markers and QTLs. Among these tested models, GAPIT performed better than FARMCPU and TASSEL when considering the obtained QQ plot based on *p*-values (Appendix A).

Nine common markers for all three methods for five measured traits were identified in the present study. No common markers for the three methods were identified on chromosomes 1H and 6H. In the present study, the marker “JHI-Hv50k-2016-435708” was associated with relative plant height and relative thousand-grain weight on chromosome 7H. These two traits are controlled by several genes and are positively correlated [41]. He et al. [45] reported eight markers on barley chromosomes 2H and 5H that are associated with plant height and thousand-grain weight. 

The identified common markers (among all three methods, Table 4) were screened for candidate genes according to published functional gene annotations of Morex V2 [28], leading to the identification of three high-confidence genes on chromosome 2H (BOPA2_12_21049, JHI-Hv50k-2016-123144, and JHI-Hv50k-2016-142550) and one high-confidence gene on chromosome 4H (BOPA1_2955-452), respectively. As a potential candidate resistance gene, the *Dihydrofolate reductase (DHFR)* gene on chromosome 2H was found to be colocalized with the “BOPA2_12_21049” marker, which was associated with relative plant height. This gene plays several important roles in cell metabolism, catalyzes the conversion of dihydrofolate to tetrahydrofolates (synthesis of 5,6,7,8-tetrahydrofolate) [46,47], and may lead to tolerance through compensation of virus-induced metabolic changes in its host. A second high-confidence gene “*NBS-LRR disease resistance*” was identified on chromosome 2H and was co-located with the identified marker “JHI-Hv50k-2016-123144” that is associated with the relative number of ears per plant. This gene belongs to a large group of disease resistance genes (*R* genes), which are involved exclusively in a non-membrane-bound form in qualitative resistance to different viruses in various host plants [48]. In addition to the two identified candidate genes on chromosome 2H, *Dihydroflavonol 4-reductase* was identified as a third gene at 491 bp distance from marker “JHI-Hv50k-2016-142550”. This gene plays a role in flavonoid metabolism. It is involved in the production of anthocyanins and proanthocyanidins [49]. Flavonoids have been shown to have antiviral activity [50]. The identified marker on chromosome 4H corresponds to a gene coding for a cysteine proteinase inhibitor located at a distance of 374 bp from the “BOPA1_2955-452” marker. Cysteine proteinase inhibitors were reported to increase plant resistance against pathogens and insects [51,52,53,54]. The increase in resistance to potyviruses using cysteine proteinase inhibitors in transgenic tobacco plants was reported by Gutierrez-Campos and Torres-Acosta [48]. Furthermore, Carrillo et al. [54] indicated that the barley cysteine-proteinase inhibitor reduced the performance of two aphid species in artificial diets and transgenic *Arabidopsis thaliana* plants.

The identification of WDV-resistant or -tolerant genotypes, as well as an understanding of the genetic background of plants, is a prerequisite to reduce the negative effects of this virus on plant production. In this context, identified markers or QTLs not only provide a relevant genetic basis for breeding but also enhance our knowledge about genomic regions, which control WDV resistance in barley. 

## 5. Conclusions

We present a first GWAS study using the 50K iSelect chip for barley to identify associated markers for resistance/tolerance to WDV. In this study, three different statistical models (MLM, CMLM, and FARM CPU) were used to validate the results and identify significant marker–trait associations. On the basis of phenotypic data, three genotypes were defined as resistant, and 29 genotypes were defined as tolerant. GWAS revealed nine markers significantly associated with resistance/tolerance to WDV. The development of KASP markers (Kompetitive Allele Specific PCR) based on the obtained common significant markers could be a valuable tool for plant breeding and replace the classical screening method including vector insects and viruses. 

## Figures and Tables

**Figure 1 viruses-15-01568-f001:**
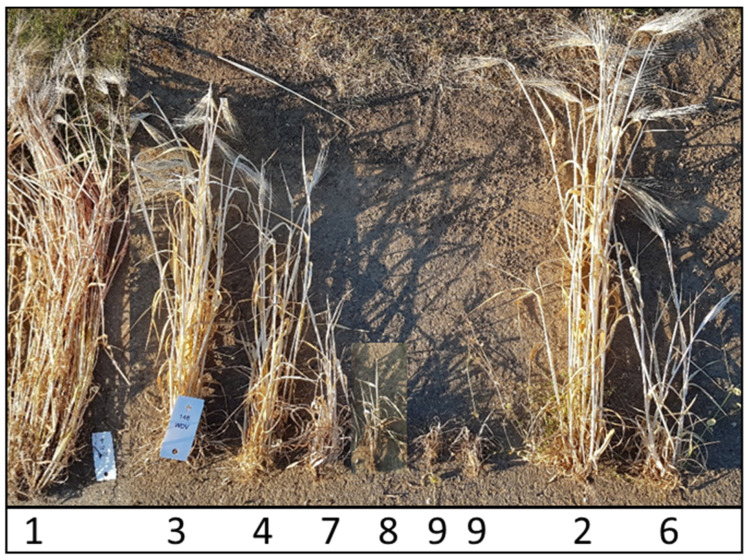
Symptom scoring scale 1–9: 1, free of symptoms; 2, minimally dwarfed; 3, weak growth reduction; 4, weak growth reduction and reduced number of ears per plant; 5, moderately dwarfed and reduction in tillers and ears (it is not shown); 6, moderately to severely dwarfed and few ears; 7, severely dwarfed; 8, heavily dwarfed; 9, dead plant and no yield.

**Figure 2 viruses-15-01568-f002:**
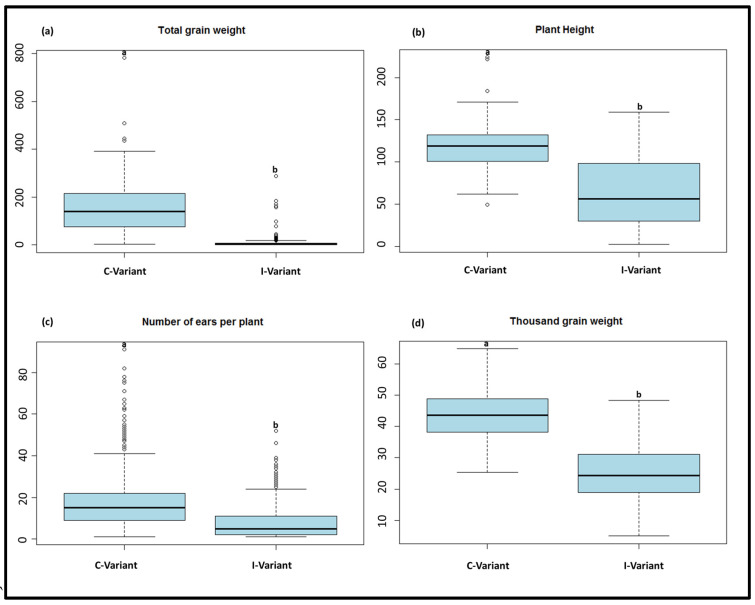
Boxplot based on genotype means for (**a**) total grain weight, (**b**) plant height, (**c**) number of ears per plant and (**d**) thousand grain weight. Mean values with different letters (a and b) indicate significant (*p* < 0.001) differences between C and I-variant.

**Figure 3 viruses-15-01568-f003:**
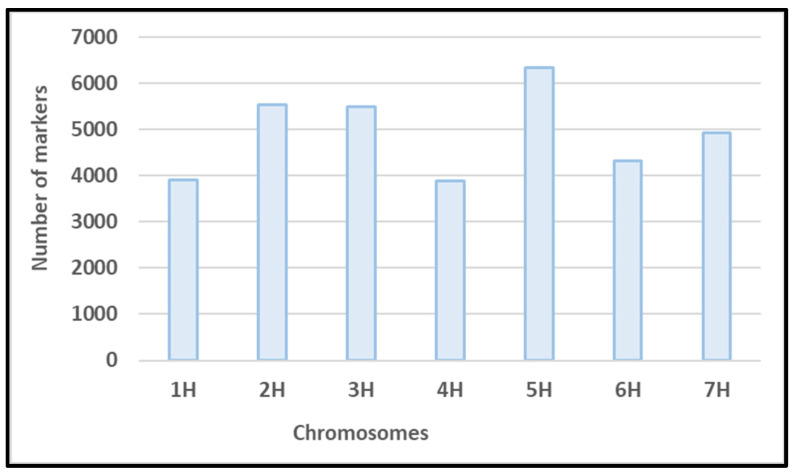
Distribution of SNP markers across all seven barley chromosomes.

**Figure 4 viruses-15-01568-f004:**
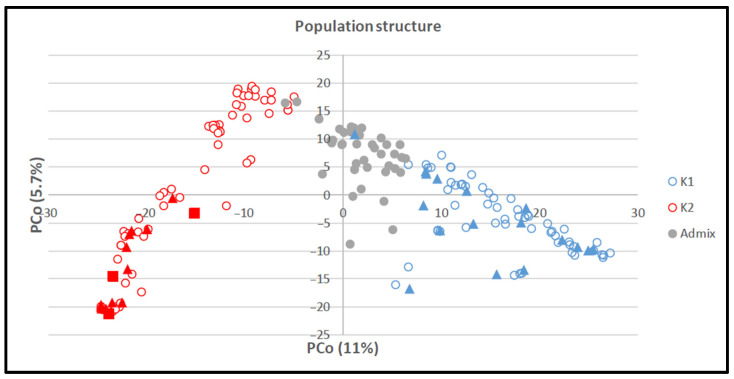
Principal coordinate analysis (PCoA) according to Structure grouping of 191 barley genotypes. Legend: blue empty dots, genotypes assigned to Structure group 1; red empty dots, genotypes assigned to Structure group 2; gray dots, genotypes assigned to the admixed group. The rectangles and triangles indicate resistant and tolerant genotypes, respectively.

**Figure 5 viruses-15-01568-f005:**
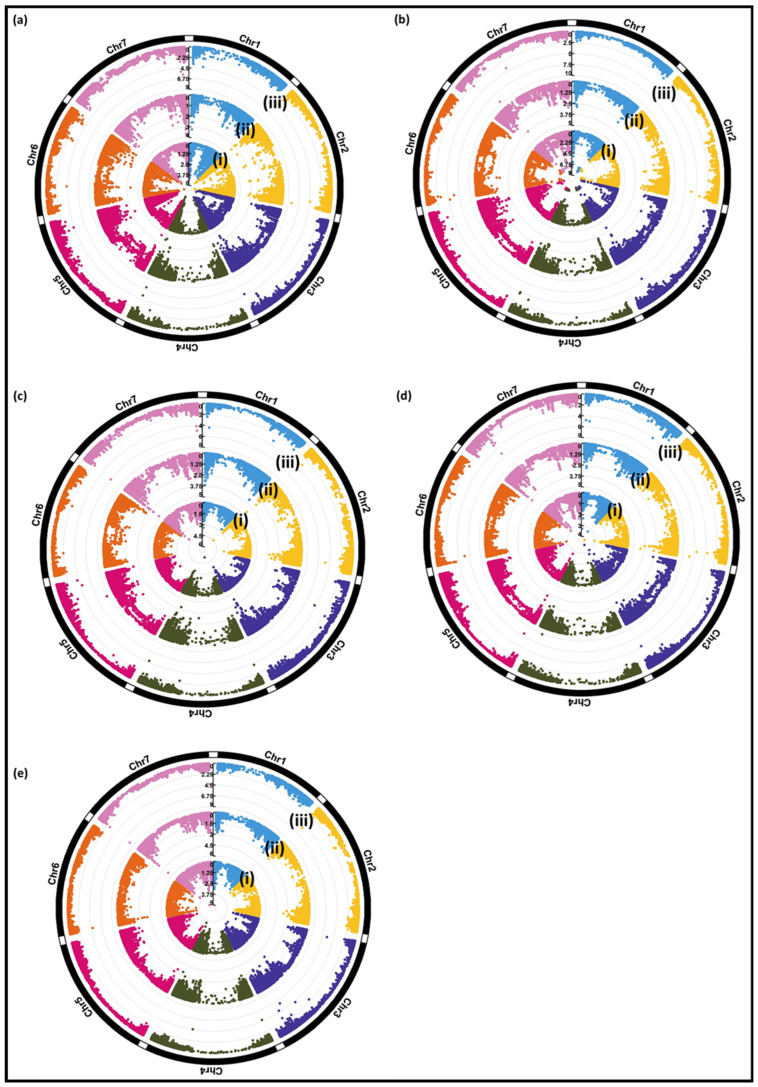
Circular Manhattan plot indicating three different GWAS methods: (i) MLM (in TASSEL), (ii) CMLM (in GAPIT), and (iii) FarmCPU for (**a**) relative virus titer, (**b**) relative total grain weight (**c**) relative plant height, (**d**) relative number of ears per plant, and (**e**) relative thousand-grain weight.

**Table 1 viruses-15-01568-t001:** List of evaluated traits under two different variants.

Trait	Abbreviation	Method of Measurement	Unit
Relative virus titer	ELISA-60	Extinction value of a double antibody sandwich enzyme-linked immunosorbent assay (DAS-ELISA)	
Total grain weight	ToGW	Weight all harvested seeds per plant	g
Plant height	HEI	Measure plant length from basis to top of the head	cm
Number of ears per plant	NEP	Count number of ears after harvesting	
Thousand grain weight	TGW	Weigh 1000 grains after threshing	g

**Table 2 viruses-15-01568-t002:** Trait performance of 191 investigated barley genotypes.

Trait	Treatment ^a^	N ^b^	Mean ^c^	Minimum ^d^	Maximum ^d^	Sd ^e^	CV ^f^
Relative virus titer	I-variant	1866	0.3	−0.03	1.84	0.6	179.7
Total grain weight	I-variant	209	11.3	0	287.4	31.7	280.7
	C-variant	229	156.3	2.2	781.7	102.6	65.7
Plant height	I-variant	520	62.9	1	159	40.4	64.2
	C-variant	1163	117.5	49	224	20.3	17.2
Number of ears per plant	I-variant	524	6.9	0	52	8.2	118.3
	C-variant	1160	18.4	1	117	12.1	65.8
Thousand grain weight	I-variant	208	25.5	5	48.3	8.7	34.1
	C-variant	229	43.5	25.37	64.8	7.8	17.9

^a^ Treatment: infected (I-variant) and control (C-variant). ^b^ Number of observations. ^c^ Mean value. ^d^ Maximum and minimum. ^e^ Standard deviation. ^f^ Coefficient of variation (in %).

**Table 3 viruses-15-01568-t003:** ANOVA result of all five measured traits for 191 investigated wheat genotypes.

Trait	Effect	Degrees of Freedom	F-Value	Pr > F
Relative virus titer	Genotype (G)	190	2.26	<0.0001
Total grain weight	Genotype (G)	190	1.05	0.49
	Treatment (T)	1	364.08	<0.0001
	G × T	184	0.84	0.74
Plant height	Genotype (G)	190	5.43	<0.0001
	Treatment (T)	1	3651.82	<0.0001
	G × T	186	9.26	<0.0001
Number of ears per plant	Genotype (G)	190	1.04	0.50
	Treatment (T)	1	443.86	<0.0001
	G × T	186	1.83	<0.0001
Thousand grain weight	Genotype (G)	190	3.05	<0.0001
	Treatment (T)	1	1233.26	<0.0001
	G × T	184	2.19	<0.0001

**Table 4 viruses-15-01568-t004:** List of commonly identified significant associated trait markers through three different used methods.

Trait	Marker Name	Chr ^a^	Pos ^b^	*p* Value			Identified Genes in QTL Region
				Gapit ^c^	Tassel ^c^	FarmCPU	
Relative virus titer	JHI−Hv50k−2016−202912	3H	562,758,917	3.3 × 10^−4^	2.6 × 10^−4^	2.2 × 10^−4^	
Relative total grain weight	JHI−Hv50k−2016−196649	3H	534,052,013	6.8 × 10^−5^	7.7 × 10^−4^	2.8× 10^−6^	
Relative total grain weight	BOPA1_2955−452	4H	552,300,974	9.5 × 10^−4^	2.4 × 10^−5^	9.4 × 10^−5^	*Cysteine proteinase inhibitor*
Relative total grain weight	BOPA2_12_10333	5H	554,416,618	3.4 × 10^−4^	1.1 × 10^−4^	1.7 × 10^−4^	
Relative plant height	BOPA2_12_21049	2H	31,329,721	3.3 × 10^−5^	3.5 × 10^−5^	6.6 × 10^−4^	*Dihydrofolate reductase*
Relative plant height	JHI−Hv50k−2016−435708	7H	1,402,273	1.5 × 10^−5^	7.4 × 10^−5^	1.1 × 10^−6^	
Relative number of ears per plant	JHI−Hv50k−2016−123144	2H	631,278,948	1.8 × 10^−4^	8 × 10^−4^	2.7 × 10^−6^	*NBS−LRR disease resistance protein*
Relative number of ears per plant	JHI−Hv50k−2016−142550	2H	666,139,797	6.4 × 10^−5^	1.0 × 10^−4^	1.7 × 10^−7^	*Dihydroflavonol 4−reductase*
Relative thousand grain weight	JHI−Hv50k−2016−435708	7H	1402273	4.8 × 10^−6^	1.7 × 10^−5^	8.7 × 10^−9^	

^a^ Chromosome. ^b^ Position. ^c^ CMLM was applied in GAPIT and TASSEL.

## Data Availability

The data presented in this study are openly available in OpenAgrar at doi.org/10.5073/20230616-101640-0, [55].

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
