# Peer review of "Identification of Markers Associated with Wheat Dwarf Virus (WDV) Tolerance/Resistance in Barley (Hordeum vulgare ssp. vulgare) Using Genome-Wide Association Studies"

_viruses, 2023, doi:10.3390/v15071568_

Round 1

Reviewer 1 Report

The finding by GWAS of more than 100 markers of barley resistance to WDV seems to be a valuable achievement, but relating these markers to a select few symptoms of a single disease would only make sense if the Authors showed that these symptoms appear, at least in selected genotypes independently of each other.

·         In this case, I would suggest that Table 1 of the supplement should be supplemented with ELISA results, as well as the values of other measured symptoms (features) for infected and non-infected plants.

·         Table 2. does not need to be repeated in supplementary data.

·         There is no sequence information about the WDV virus under study – it is a small  ssDNA molecule, which should be cloned, sequenced and its nucleotide sequence should be available for the readers.

·         Since phenotyping was performed visually, the photographs of examples of infected plants with streaks should be shown, or a link to documented representative group of plants tested, healthy and infected, those more or less susceptible and more or less resistant.

Minor remarks:

A more detailed description of infection would be helpful for thee readers (line 95):

“To increase the infection pressure, one WDV-infected barley plant was placed in a short distance between each row of the 96 inoculated variant.”

(DAS-ELISA) (line 104) a citation seems to be missing.

Could it be explained shortly why the extent of leaf chlorosis upon infection was excluded form evaluated symptoms?

Author Response

Dear Reviewer 1,

Thanks.

Reviewer 2 Report

The manuscript titled 'Genome-wide association studies for the identification of markers associated with Wheat dwarf virus (WDV) tolerance/resistance in barley (Hordeum vulgare ssp. vulgare)' by Soleimani et al appears to be an intriguing study. The authors may consider refining the title by revising the use of 'tolerance/resistance' to make it more concise and precise.

In the abstract, it might be advisable to avoid explicitly mentioning all the software tools used, such as Tassel, GAPIT, and FARM CPU. Additionally, the authors should provide a justification for selecting a particular software tool for the GWAS analysis, as the use of redundant software may not contribute to better results.

In the introduction section, the phrase 'High dense marker sets' could be rephrased to convey the same meaning more effectively.

In the same section, it would be beneficial to clarify the distinction between tolerance mechanisms and resistance mechanisms. Furthermore, it is worth mentioning that no QTL regions associated with tolerance or resistance to WDV have been identified in barley so far, whereas they have been recently identified in wheat. Therefore, the identification of QTL regions and the development of diagnostic markers related to tolerance or resistance to WDV in barley are of great importance.

In the materials and methods section, it is recommended to have separate subsections for Phenotyping and Genotyping to provide a clearer structure for the methodology.

Additionally, it would be valuable if the authors could provide the genotyping data as a supplementary file, enhancing the reproducibility and accessibility of their study.

The overall quality of English is good. Please note I am not native English speaker 

Author Response

Dear Reviewer 2,

Thanks.

Round 2

Reviewer 1 Report

The authors performed GWAS analysis of 190 barley genotypes using 34,408 SNPs and to eliminate false positives, used 3 statistical models to select the most likely genes responsible for the resistance of barley varieties to WDV . The results of this work can be very useful in breeding / selecting barley cultivars that show resistance to WDV, to limit the use of insecticides in agriculture. 

Reviewer 2 Report

The revised version of the MS looks appropriate.